# The Effect of the COVID-19 Lockdown on the Position-Specific Match Running Performance of Professional Football Players; Preliminary Observational Study

**DOI:** 10.3390/ijerph182212221

**Published:** 2021-11-21

**Authors:** Damir Sekulic, Sime Versic, Andrew Decelis, Jose Castro-Piñero, Dejan Javorac, Goran Dimitric, Kemal Idrizovic, Igor Jukic, Toni Modric

**Affiliations:** 1Faculty of Kinesiology, University of Split, 21000 Split, Croatia; dado@kifst.hr (D.S.); sime.versic@kifst.hr (S.V.); 2HNK Hajduk, 21000 Split, Croatia; 3Institute for Physical Education and Sport, Faculty of Education, University of Malta, MSD 2080 Msida, Malta; andrew.decelis@um.edu.mt; 4GALENO Research Group, Department of Physical Education, Faculty of Education Sciences, University of Cádiz, Avenida República Saharaui s/n, 11519 Puerto Real, Spain; jose.castro@uca.es; 5Instituto de Investigación e Innovación Biomédica de Cádiz (INiBICA), 11009 Cádiz, Spain; 6Faculty of Sport and Physical Education, University of Novi Sad, 21000 Novi Sad, Serbia; javorac.dejan@gmail.com (D.J.); dimitrg@gmail.com (G.D.); 7Faculty for Sport and Physical Education, University of Montenegro, 81400 Niksic, Montenegro; kemo@t-com.me; 8Faculty of Kinesiology, University of Zagreb, 10000 Zagreb, Croatia; igor.jukic@kif.hr

**Keywords:** SARS-CoV-2, match running performance, soccer, match demands

## Abstract

The COVID-19 pandemic interrupted professional football in the 2019/2020 season, and football experts anticipate that the consequences of lockdown measures will negatively affect the physical performance of players once competition restarts. This study aimed to evaluate position-specific match running performance (MRP) to determine the effect of COVID-19 lockdowns on the physical performance of professional football players. Players’ MRPs (*n* = 124) were observed in matches before and after the COVID-19 lockdown in the 2019/2020 season of the highest level of national competition in Croatia and were classified according to player position: central defenders (CD; *n* = 42), fullbacks (FB; *n* = 20), midfielders (MF; *n* = 46), and forwards (FW; *n* = 16). The MRPs were measured using Global Positioning System, and included the total distance covered, low-intensity running (≤14.3 km/h), running (14.4–19.7 km/h), high-intensity running (≥19.8 km/h), total accelerations (>0.5 m/s^2^), high-intensity accelerations (>3 m/s^2^), total decelerations (less than –0.5 m/s^2^), and high-intensity decelerations (less than –3 m/s^2^). The results indicated that, in matches after the COVID-19 lockdown, (i) CDs and FBs featured lower running and high-intensity running (t-value: from 2.05 to 3.51; all *p* < 0.05; moderate to large effect sizes), (ii) MFs covered a greater distance in low-intensity running and achieved a lower number of total accelerations, and total and high-intensity decelerations (t-value: from –3.54 to 2.46; all *p* < 0.05, moderate to large effect sizes), and (iii) FWs featured lower high-intensity running (t-value = 2.66, *p* = 0.02, large effect size). These findings demonstrate that the physical performances of football players from the Croatian first division significantly decreased in matches after the COVID-19 lockdown. A combination of inadequate adaptation to football-specific match demands and a crowded schedule after the competition was restarted most likely resulted in such an effect.

## 1. Introduction

At the end of 2019, the COVID-19 virus appeared and rapidly spread worldwide, which led to the declaration of a pandemic on 11 March 2020 [1]. Countries around the globe enacted different measures for stopping or slowing down the spread of the COVID-19 disease. The major strategy was the implementation of social distancing measures and lockdown (home confinement), which included closures of all places where a large number of people can gather (schools, universities, churches, training facilities, and sports clubs) [2]. Athletes were not exempted from the abovementioned rules, and thus, the majority of competitions were cancelled.

In Croatia, the last official football match of the highest national football competition was played on 9 March 2020. During the COVID-19 lockdown period, football players were unable to carry out their professional work in the usual way. Specifically, due to the general medical rules which suggested that physical distancing of two meters between players had to be maintained to prevent saliva droplets from coming into contact with other players [3], typical football team trainings were not conducting until the beginning of May. Subsequently, football players experienced training protocols (i.e., training with no collaboration or opposition) that were completely different from competition conditions for about two months [4]. Since matches resumed from 5 June, players have been able to engage in football-specific work for only 3–4 weeks. In general, such short period of involvement in football-specific work could limit players’ adaptations to the match demands. Indeed, football experts anticipated that this may negatively affect the physical performance of players once competition restarts [3].

Physical performance in football is commonly analyzed by quantifying the match running performance (MRP), which includes the total distance covered, distances covered in various speed zones, and acceleration rates [5,6,7,8,9,10]. For example, it is well known that football players can cover between 9 and 14 km, performing 0.7–3.9 km of high-speed distance and 0.2–0.6 km of sprint distance, with ~600 accelerations, during matches [11,12,13,14,15]. Previous studies demonstrated that MRP varies according to the different playing positions of the players due to the different tactical roles in the matches [16,17]. Specifically, central midfielders cover the highest overall distance during the matches, while wingers and fullbacks cover the greatest distance in terms of high intensity [9,18]. To enable appropriate adaptation to such requirements, an adequate period of training is essential [19,20]. However, in the 2019/2020 season, this was interrupted by the COVID-19 lockdown.

Although many hypotheses have been proposed regarding the possible negative impacts the COVID-19 lockdown will have had on the physical performance of players once competition is restarted [3,4,21], recent studies have provided inconsistent findings. In brief, Radzimiński et al. recently reported that the physical performance in German Bundesliga was not affected by the COVID-19 lockdown, while on the other hand, Polish Ekstraklasa matches after the COVID-19 lockdown had a significantly shorter total and high-intensity running distance [22]. García-Aliaga et al. compared the MRPs before and after the COVID-19 lockdown in Spain and found lower medium-speed running (14.1–21 km/h), high-speed running (21.1–24 km/h), and sprinting distances (>24 km/h) in matches after the COVID-19 lockdown. The authors suggested that such a difference could have been caused by different break lengths and different restrictions implemented in these countries during the pandemic lockdown [22].

These two studies provided valuable findings about the effect of the COVID-19 lockdowns on MRP. However, knowledge about this issue is still limited due to the lack of studies specifically considering the effects of the COVID-19 lockdowns on MRP in various environments. Indeed, Brito de Souza et al. emphasized the necessity of investigating the effect of the COVID-19 lockdowns on performance in professional football players from different leagues [23]. Furthermore, Radzimiński et al. suggested that an analysis of individual changes according to playing position should also be carried out in further studies [22]. However, to the best of our knowledge, no study has analyzed the effect of the COVID-19 pandemic on position-specific MRP. Considering the fact that there has been no study to investigate the effect of COVID-19 on Croatian football players, we were of the opinion that further evidence may improve our understanding of this issue. Therefore, the main objective of this study was to evaluate the position-specific MRP in the highest national football competition in Croatia before and after the COVID-19 lockdown, in order to determine the effect of the COVID-19 lockdown on the physical performance of professional football players. Initially, we hypothesized that MRP would be lower after the COVID-19 lockdown, with certain position-specific differences.

## 2. Materials and Methods

### 2.1. Participants and Design

Twenty-one professional football players (M ± SD, age 24.19 ± 2.46, body mass 77.32 ± 4.45, height 182.32 ± 6.32) from the same team participated in this study, and all signed an informed consent form agreeing to participate in the study. The players were classified according to their position in the game as central defenders (CD), fullbacks (FB), midfielders (MF), and forwards (FW) (i.e., the team played in the tactical formation 3–5–2, which does not include wingers).

The players’ MRPs were analyzed during all matches (*n* = 17) from the second half of the Croatian highest national football competition in the 2019/2020 season. Seven matches were played before the COVID-19 lockdown, and 10 matches after. Matches from the first half of the season were not included due to the winter break, during which players do not participate in team training programs (the Croatian First Football League season is split into two halves, with a winter break of approximately 20 days between each half). The MRPs were obtained with GPS (see later for details) and used as cases in this study. For methodological reasons, only players who played a whole match were analyzed. A total of 121 MRPs were obtained and divided according to player position into the four groups: CD = 38, FB = 20, MF = 46, and FW = 16. Position-specific MRPs were later divided into the two groups, before and after COVID-19 lockdown: CD before = 19/CD after = 19; FB before = 8/FB after = 12; MF before = 16/MF after = 30; FW before = 5/FW after = 11. To determine the effect of the COVID-19 lockdown, position-specific MRPs before and after the COVID-19 lockdown were paired for comparison.

The timeline of the Croatian First Division 2019/2020 season is presented in Figure 1 below. The season under study was suspended on 9 March 2020, and players trained at home until 13 April. Home-based training typically aimed to improve strength and endurance by applying basic conditioning exercises. During the following two weeks, players were included in individual field-based work, which typically consisted of running and technical drills. From 27 April, players started to work in small groups of a maximum of five players. Training programs were designed to respect 2 m physical distancing, and mostly included running and technical drills. This phase lasted 2 weeks and did not include football-specific work, such as small-sided games. On 11 May, players started normal team training, including small- and large-sided games. After three weeks of this phase, one classic microcycle was applied in order to prepare the team for the competition period (i.e., applying tapering strategies). The first post-lockdown match was played on 5 June (Figure 1).

### 2.2. Procedure

Players’ MRPs were measured using GPS devices (Vector S7, Catapult, Catapult Sports Ltd., Melbourne, Victoria, Australia) with a sampling frequency of 10 Hz. Each player wore the same GPS device in all matches in order to avoid interunit variability. These devices have already been investigated for metrics and were found to be appropriately reliable and valid in sports settings (i.e., less than 1% measurement error, and 80% of common variance with running speed measured by timing gates) [9,24,25]. This tracking system was already used in many other studies [26,27].

The MRPs included the following variables: total distance covered (m); distance in different speed categories (low-intensity running (≤14.3 km/h), running (14.4–19.7 km/h), high-intensity running (≥19.8 km/h)); number (frequency) of total accelerations (>0.5 m/s^2^); number of high-intensity accelerations (>3 m/s^2^); number of total decelerations (less than –0.5 m/s^2^), and number of high-intensity decelerations (less than –3 m/s^2^).

### 2.3. Data Analysis

Levene’s test was used to test for the equality of variances. The normality of the distributions was checked by the Kolmogorov–Smirnov test, which confirmed that all variables were normally distributed (K-S *p* > 0.05).

The descriptive statistics were presented as the means ± standard deviations. T-test for independent samples were used to evaluate differences in MRPs before and after the COVID-19 lockdown. Calculations were conducted separately for each position.

Cohen’s d effect size was calculated in order to compare the magnitude of the differences between groups for certain variables, and interpreted as trivial (d < 0.20), small (0.20 ≤ d < 0.50), medium (0.50 ≤ d < 0.80), and large (d > 0.80) [28].

For all analyses, Statistica 13.0 (TIBCO Software Inc., Greenwood Village, CO, USA) was used, and *p* < 0.05 was applied.

## 3. Results

Table 1, below, presents the descriptive parameters and differences in total distance, low-intensity running, running, and high-intensity running before and after the COVID-19 lockdown. CDs’ running (t-value = 2.82, *p* = 0.01, large effect size) and high-intensity running (t-value = 2.05, *p* = 0.04, moderate effect size) were significantly lower in matches after the COVID-19 lockdown. MFs covered a greater distance in low-intensity running (t-value = −3.54, *p* = 0.01, large effect size), while FBs showed greater running (t-value = −3.51, *p* = 0.01, large effect size) and high-intensity running (t-value = 3.11, *p* = 0.01, large effect size) in matches after the COVID-19 lockdown. FWs’ high-intensity running was significantly lower in matches after the COVID-19 lockdown (t-value = 2.66, *p* = 0.02, large effect size).

Table 2, below, presents the descriptive parameters and differences in total and high-intensity accelerations, and total and high-intensity decelerations before and after the COVID-19 lockdown. FBs showed a significantly lower number of high-intensity decelerations (t-value = 3.05, *p* = 0.01, large effect size) in matches after the COVID-19 lockdown. MFs performed lower total accelerations (t-value = 2.35, *p* = 0.02, moderate effect size) and decelerations (t-value = 2.42, *p* = 0.02, moderate effect size), and high-intensity decelerations (t-value = 2.46, *p* = 0.02, moderate effect size) in matches after the COVID-19 lockdown.

## 4. Discussion

The main purpose of this study was to evaluate position-specific MRPs before and after the COVID-19 lockdown in the highest national football competition in Croatia. The results indicated a significant decrease in the physical performance in matches after the COVID-19 lockdown for players in all playing positions.

Supporting the results from the previous studies [4,22], the findings from this study demonstrated that CDs, FBs, MFs, and FWs played at a lower game pace in matches after the COVID-19 lockdown. Specifically, we found that CDs and FBs covered 15–20% less distance in the running zone (i.e., moderate-intensity running) in matches after the COVID-19 lockdown (large effect sizes). Additionally, CDs, FBs, and FWs engaged in 20–25% less high-intensity running (moderate to large effect sizes), indicating that these players played at a lower game pace in matches after the COVID-19 lockdown. Although we did not find significant differences in high-intensity running for MFs, the descriptive parameters indicated a ~10% lower high-intensity running distance after the COVID-19 lockdown. Moreover, MFs’ greater engagement in low-intensity running after the COVID-19 lockdown (large effect size) indicated that MFs played at a lower game pace (i.e., greater walking and jogging distance, and lower high-intensity distance), similarly to the other players. Furthermore, MFs carried out fewer total accelerations and decelerations, and high-intensity decelerations, in matches after the COVID-19 lockdown (moderate effect sizes), indicating that MFs were less active in matches after the COVID-19 lockdown.

Although these findings clearly indicate reduced match intensity for all playing positions after COVID-19 lockdown, some specificities in changes of MRP should be noted. Most importantly, it seems that MRP changed differently due to the different game duties of players on different playing positions. For example, for CDs, the largest differences between MRP before and after COVID-19 lockdown matches were found for distance covered in running zone (i.e., large effect size). Since most of CD’s efforts in the matches are performed in the zone of running (14.4–19.7 km/h) [9], such findings are actually expected. On the other hand, the FBs’ main technical requirements are the number of entries to the third part of the pitch (i.e., pressing) and the number of crosses [29,30], which are usually performed at moderate and higher speeds. Not surprisingly, for this playing position (FBs), we evidenced the largest differences for running and high-intensity running (both large effect sizes). Next, the sprint distance covered is an important determinant of FWs’ overall game performance [15]. Interestingly, the largest differences between MRP before and after COVID-19 lockdown matches for FWs were evidenced for high-intensity running (large effect size) which includes high-speed running (19.8–25.1 km/h) and sprinting (>25.2 km/h). The main role of MFs is to organize the offense by proper ball control and passes, rather than by invasion into the opponent’s area [29,31]. Therefore, MFs’ game duties are most likely related to the accelerations and decelerations. Similar to other playing positions, MFs’ MRPs that are related to their main game duties (e.g., total accelerations/decelerations and high-intensity decelerations) were decreased in matches after COVID-19 lockdown.

Previous studies which demonstrated similar findings speculated on the reasons for this decline in physical performance [4,22]. In brief, Radzimiński et al. reported that Polish Ekstraklasa matches after COVID-19 lockdown included significantly shorter total and high-intensity running times. Authors reported that the reduced MRPs could be due to changes in physical fitness and body composition [22]. On the other hand, García-Aliaga et al. found lower distances covered at medium, high, and sprinting speeds in matches after the COVID-19 lockdown, and considered that the decrease in the physical performance may have been caused by a crowded competition schedule (i.e., after competitions were restarted, national federations scheduled teams to play matches more often) [4]. Since the authors of this study were deeply involved in the training process of the players, we believe that the true reasons for the decrease in physical performance may be somewhere in the middle.

Giving further detail, during the period of ~8 weeks without team training, Croatian players in the first phase (i.e., when it was prohibited to train outside) engaged in four weeks of home-based training programs. In the second phase, when training outside was allowed with physical distancing, the players trained for two weeks individually and two weeks in small groups on the standard training field. In this period, an individual approach was applied, meaning that players worked individually with fitness and assistant coaches according to their current fitness status [32]. The focus was primarily on endurance and strength/power capacities, which undoubtedly decreased during the pandemic quarantine [3,33]

As suggested previously, players started with light aerobic work, which was gradually increased to high-intensity anaerobic work [3]. As a consequence, when players returned to the team trainings (i.e., after four weeks), their fitness status was reasonable. Given that they did not undertake specific football work, in this stage, they were not adequately adapted to match demands. To improve this, in the next four weeks, team football programs were applied. However, as suggested by international football experts, this work was added gradually to avoid injuries [3]. This means that during the first week, players were not maximally loaded. Additionally, in the last phase (i.e., before the first post-lockdown match), tapering strategies were applied in order to maximize players’ performances before the restart of the competition [34,35,36].

Consequently, although the players participated in four weeks of team programs, they most likely did not fully accomplish adaptation to match demands. On top of this, when the competition was restarted, the national federations scheduled the matches to be played more often, with fewer days between them [4]. From our experience, this crowded schedule reduced the time for recovery. Meanwhile, due to some players being infected with COVID-19, rotating of the players in matches was limited. Taken together, the decrease in the players’ physical performances (i.e., playing at a lower game pace) in matches after the COVID-19 lockdown was most likely a result of a combination of inadequate adaptation to football-specific match demands and a crowded schedule after the competition was restarted.

### Limitations and Strengths

Although this study emphasizes possible consequences derived from COVID-19 confinement and its implications for professional football players’ performance, some limitations should be considered. The main limitation of this study was the fact that we did not observe contextual factors. Specifically, factors such as match location, match outcome, strength of opponent, or team formation have all been demonstrated as important determinants of physical performance in football [8,31,37,38]. Hence, we are fully aware that such factors could have influenced the results of the presented analysis and that, consequently, conclusions drawn in this study should be cautiously interpreted. In addition, it must be emphasized that only one team was observed, and therefore findings from this study cannot be generalized. However, this is a very common obstacle in studies involving professional and elite players [39,40]. Future studies should address these limitations by including other teams and contextual factors to provide more comprehensive understanding effect of the COVID-19 lockdown on MRP.

On the other hand, this study has several strengths. This is the first study which analyzed the effect of the COVID-19 lockdown on position-specific MRPs. In addition, findings from this study were provided by the authors that were included in the training process of football players during the COVID-19 lockdown, and therefore valuable insights into coping with the COVID-19 pandemic in football were obtained. Finally, since the world is still battling the COVID-19 pandemic, the results of this study may help football and fitness coaches face possible new lockdown measures in better ways.

## 5. Conclusions

This study has shown that the physical performances of football players from the Croatian first division significantly decreased in matches after the COVID-19 lockdown. Specifically, CDs’ and FBs’ running and high-intensity running was lower, MFs covered greater distance in low-intensity running and featured lower acceleration rates, while FWs experienced lower high-intensity running. The long period without football-specific team trainings during the COVID-19 lockdown and a crowded match schedule after the competition resumed most likely had an effect on players’ physical performances in matches after the COVID-19 lockdown. Since these factors could not be managed by coaching staff, rotations in the starting line-ups and making suitable substitutions may be useful strategies to maintain physical performance during matches.

## Figures and Tables

**Figure 1 ijerph-18-12221-f001:**
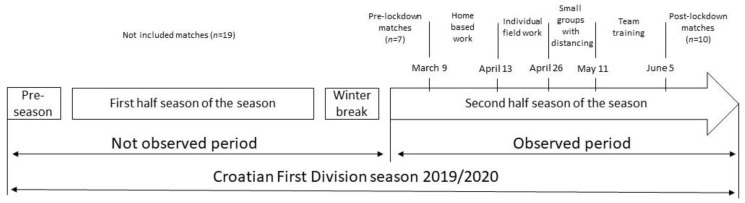
Timeline of the Croatian First Division 2019/2020 season.

**Table 1 ijerph-18-12221-t001:** Descriptive statistics and differences in total distance, low-intensity running, running, and high-intensity running before and after lockdown in Croatia.

Playing Position	Title	Total Distance (m)	Low-Intensity Running (m)	Running (m)	High-Intensity Running (m)
Central Defenders	Before	10,247 ± 643	8323 ± 439	1353 ± 256	571 ± 217
	After	10,211 ± 541	8602 ± 486	1157 ± 161	453 ± 130
	T-value (*p*)	0.19 (0.85)	−1.85 (0.07)	2.82 (0.01)	2.05 (0.04)
	Effect size	0.06	−0.62	0.94	0.68
Full Backs	Before	11,106 ± 430	8269 ± 404	1819 ± 259	1018 ± 162
	After	10,549 ± 668	8316 ± 439	1442 ± 220	790 ± 160
	T-value (*p*)	2.08 (0.06)	−0.24 (0.81)	3.51 (0.01)	3.11 (0.01)
	Effect size	0.98	−0.11	1.65	1.46
Midfielders	Before	11,492 ± 478	8640 ± 318	2116 ± 207	735 ± 197
	After	11,668 ± 625	9063 ± 416	1937 ± 352	669 ± 181
	T-value (*p*)	−0.98 (0.33)	−3.54 (0.01)	1.87 (0.07)	1.15 (0.25)
	Effect size	−0.30	−1.07	0.57	0.35
Forwards	Before	10,670 ± 605	7968 ± 274	1651 ± 270	1051 ± 98
	After	10,392 ± 912	7875 ± 506	1703 ± 460	814 ± 185
	T-value (*p*)	0.62 (0.55)	0.38 (0.71)	−0.23 (0.82)	2.66 (0.02)
	Effect size	0.33	0.20	−0.12	1.42

**Table 2 ijerph-18-12221-t002:** Descriptive statistics and differences in total and high-intensity accelerations, and total and high-intensity decelerations before and after lockdown in Croatia.

Playing Position	Title	Total Accelerations (Count)	Total Decelerations (Count)	High-Intensity Accelerations (Count)	High-Intensity Decelerations (Count)
Central Defenders	Before	475 ± 55	471 ± 54	22 ± 13	35 ± 13
	After	473 ± 55	474 ± 54	19 ± 4	32 ± 7
	T-value (*p*)	0.07 (0.95)	−0.12 (0.9)	0.78 (0.44)	0.75 (0.46)
	Effect size	0.02	−0.04	0.26	0.25
Full Backs	Before	491 ± 44	481 ± 41	21 ± 13.2	49 ± 10
	After	466 ± 46	457 ± 42	27 ± 6	37 ± 7
	T-value (*p*)	1.22 (0.24)	1.26 (0.22)	−1.35 (0.19)	3.05 (0.01)
	Effect size	0.58	0.60	−0.64	1.44
Midfielders	Before	517 ± 34	512 ± 32	20 ± 10	38 ± 12
	After	490 ± 39	486 ± 38	20 ± 6	31 ± 6
	T-value (*p*)	2.35 (0.02)	2.42 (0.02)	−0.07 (0.94)	2.46 (0.02)
	Effect size	0.71	0.73	−0.02	0.74
Forwards	Before	451 ± 45	441 ± 52	39 ± 18	53 ± 13
	After	445 ± 47	444 ± 47	38 ± 10	47 ± 12
	T-value (*p*)	0.26 (0.80)	−0.1 (0.92)	0.21 (0.84)	1.04 (0.31)
	Effect size	0.14	−0.06	0.11	0.56

## Data Availability

Data will be provided to all interested parties upon reasonable request.

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
