# Peer review of "The Effect of the COVID-19 Lockdown on the Position-Specific Match Running Performance of Professional Football Players; Preliminary Observational Study"

_ijerph, 2021, doi:10.3390/ijerph182212221_

Round 1
Reviewer 1 Report
For the hypothesis that there is less running after the Lockdown, none of the contextual variables that largely explain physical performance in competition have been taken into account. The effect of the score of the match has not been taken into account, and the distance traveled at high intensity is largely related to the final result of the match. For example, the centrals defenders and the fullbacks cover more distance in sprint and in high intensity distance when they lose than when they win. The wingers and forwards make more distance in sprint and high intensity when the result of the match has been victory by more than one goal. This division has not been taken into account in midfield players since they have all been considered equally and not all have the same behavior. In addition, the effect of climate or temperature is not taken into account since in summer and warmer times, the players covered less distance in sprint and high intensity since there are more stops into the match, for this we can review the distance in the World Cups or in the Euro Cups which is significantly lower. In the same way, and no less important, the effects of playing games with less recovery and tactical aspects such as a better defensive organization of the teams or the effective time that in many cases can explain these differences in physical performance in competition have not been studied. For all these reasons, the study is incomplete and it cannot be assured that the conclusions they have drawn are due to the sole effect of COVID 19.
Author Response
For the hypothesis that there is less running after the Lockdown, none of the contextual variables that largely explain physical performance in competition have been taken into account. The effect of the score of the match has not been taken into account, and the distance traveled at high intensity is largely related to the final result of the match. For example, the centrals defenders and the fullbacks cover more distance in sprint and in high intensity distance when they lose than when they win. The wingers and forwards make more distance in sprint and high intensity when the result of the match has been victory by more than one goal. This division has not been taken into account in midfield players since they have all been considered equally and not all have the same behavior. In addition, the effect of climate or temperature is not taken into account since in summer and warmer times, the players covered less distance in sprint and high intensity since there are more stops into the match, for this we can review the distance in the World Cups or in the Euro Cups which is significantly lower. In the same way, and no less important, the effects of playing games with less recovery and tactical aspects such as a better defensive organization of the teams or the effective time that in many cases can explain these differences in physical performance in competition have not been studied. For all these reasons, the study is incomplete and it cannot be assured that the conclusions they have drawn are due to the sole effect of COVID 19.
RESPONSE: Thank you very much for this criticism. No doubt, we do fully agree with your considerations that physical performance in soccer may be largely influenced by contextual factor. As you can see in “Limitations and strengths” section, we already noted this issue. Text reads: “In addition, we did not observe situational factors that may have influenced the results of the presented analysis (e.g., home advantage, match outcome, strength of opponent, playing without an audience).”
However, we must explain in detail why we conducted this study despite its possible limitations. Firstly, it is clear that soccer is multifactorial sport and that all analysed match performance (physical, technical, tactical, etc) largely dependent of plenty of variables such as the quality of the opponent, their playing style, the own playing style, the formations, match outcome, playing locaottion, level of competition, league ranking or even environmental factors (temperature, humidity, etc). On the other side, we are of opinion that is impossible to determine all factors that affect match performance in one article. Even is questionable whether is possible at all. For explanation of this we will use example of “match outcome”, but please mind that this can be applied in the same way for all other factors that you mentioned.
You stated that “the centrals defenders and the fullbacks cover more distance in sprint and in high intensity distance when they lose than when they win”. Although this may be valuable information, such conclusions are almost certainly affected by specificities of studies from which were obtained. To be more concrete I will provide specific example. In studies of Chmura at el. (2018), Andrzejewski el al. (2018) Andrzejewski el al. (2016) authors really reported that centrals defenders and the fullbacks cover more distance in sprint and in high intensity distance when they lose than when they win. But, all these studies were conducted on sample of elite German Bundesliga players. How we can be sure that previous conclusions were not affected by specificities of competition? Or by style of the play? We cannot.
Indeed, authors did not control any of situational/contextual or environmental factors such as quality of the opponent, playing style, formations, playing location, level of competition, league ranking, temperature, humidity. Even authors by itself stated this as limitation of their study with following sentence “…our findings should perhaps be applied with caution to other professional football teams and other strong European leagues, due to the different playing styles characterising different teams, as well as the specific features of different leagues.” (Chmura et al., 2021). Therefore, the information that “central defenders and the fullbacks cover more distance in sprint and in high intensity distance when they lose than when they win” cannot be treated as consensus since other factors were not controlled.
Considering that it has previously been demonstrated that physical performance are determined by many different variables such as players’ physical abilities and technical level (Modric, Versic, & Sekulic, 2021; Sæterbakken et al., 2019), team’s tactical formation (Baptista, Johansen, Figueiredo, Rebelo, & Pettersen, 2019; Modric, Versic, & Sekulic, 2020), competitive level or league ranking (Aquino et al., 2017; Bradley et al., 2013), we may also say that in studies of Chmura at el. (2018), Andrzejewski el al. (2017) and Andrzejewski el al. (2016) “it cannot be assured that the conclusions they have drawn are due to the sole effect of match outcome”. Nevertheless, all these studies were not treated as “incomplete” just because other factors were not controlled. Actually, in almost all studies which analysed physical performance in soccer you can find some factors that can influence results, but was not controlled. This is a very common obstacle, possibly due to soccer’s complex and multifactorial nature.
In general, we were aware that we did not control all factors that can affect results. On the other side, we think that study design is reasonable since many methodologies issues were undertaken to decrease influence of some factors that can affect results. For example, (i) matches from the first half of the season were not included due to the winter break, during which players do not participate in team training programs, (ii) only players who played a whole match were analysed, (iii) the team was managed by the same coaching staff before and after COVID-19 lockdown, (iv) the same tactical formation and playing style was applied in the all matches, (v) each player wore the same GPS device in all matches in order to avoid inter-unit variability, etc.
In addition, we need to say that we followed methodologies from previous studies that were published in International Journal of Environmental Research and Public Health - the same journal where we submitted our manuscript. Please see https://www.mdpi.com/1660-4601/18/7/3685/htm and https://www.mdpi.com/1660-4601/18/16/8796/htm. As you can see, these studies also did not include contextual factors and stated it as study limitation, exactly like we did in our study. After all, please mind that we characterized our study as “Brief report” and also that title of our study includes “Preliminary Observational Study”. Basically, we conducted this study in its form as “initial check” of situation in Croatia regarding effect of COVID-19 lockdown on physical performance. No doubt that extensive future investigations are needed.
Finally, despite to the discussed limitations, we really believe that our study actually has several important strengths. First, COVID-19 pandemic is global problem and we, as long-term soccer practitioners, were of opinion that is important to determine effects of lockdown (e.g., training process without standard team training) on players’ running performances on players. At the moment, pandemic is not over at other possible lockdowns may be expected. Therefore, the results of this study may help football and fitness coaches to face possible new lockdown measures in better way. Second, considering the fact that there has been no study to investigate the effect of COVID-19 on Croatian football players, we were of the opinion that is important to provide new evidences presenting situation in Croatia. Third, this is basically the first study which analysed the effect of the COVID-19 lockdown on position-specific MRPs, showing valuable knowledge how players’ running performance on different playing positions were affected. And last, probably most important, this study may encourage other authors to investigate association between running performance and COVID-19, what will enable better understanding of this issue.
In the end, to emphasize issues regarding not observed contextual factors we amended our manuscript and additionally highlighted study’s limitations in Discussion part. Text now reads: “Although this study emphasized possible consequences derived from COVID-19 confinement and its implications for professional soccer players’ performance, some limitations should be considered. The main limitation of this study was the fact that we did not observe contextual factors. Specifically, factors such as match location, match outcome, strength of opponent or team formation have all been demonstrated as important determinants of physical performance in soccer [8,31-33]. Hence, we are fully aware that such factors could have influenced the results of the presented analysis and that consequently conclusions drawn in this study should be cautiously interpreted. Also, it must be emphasized that only one team was observed, and therefore findings from this study cannot be generalized. However, this is a very common obstacle in studies involving professional and elite players [34,35]. Future studies should address these limitations by including other teams and contextual factors to provide more comprehensive understanding effect of the COVID-19 lockdown on MRP.”
References:
- Andrzejewski, M., Chmura, P., Konefał, M., Kowalczuk, E., & Chmura, J. (2017). Match outcome and sprinting activities in match play by elite German soccer players. The Journal of sports medicine and physical fitness, 58(6), 785-792.
- Andrzejewski, M., Konefał, M., Chmura, P., Kowalczuk, E., & Chmura, J. (2016). Match outcome and distances covered at various speeds in match play by elite German soccer players. International Journal of Performance Analysis in Sport, 16(3), 817-828.
- Aquino, R., Vieira, L. H. P., Carling, C., Martins, G. H., Alves, I. S., & Puggina, E. F. (2017). Effects of competitive standard, team formation and playing position on match running performance of Brazilian professional soccer players. International Journal of Performance Analysis in Sport, 17(5), 695-705.
- Baptista, I., Johansen, D., Figueiredo, P., Rebelo, A., & Pettersen, S. A. (2019). A comparison of match-physical demands between different tactical systems: 1-4-5-1 vs 1-3-5-2. PloS one, 14(4), e0214952. doi:10.1371/journal.pone.0214952
- Bradley, P. S., Carling, C., Diaz, A. G., Hood, P., Barnes, C., Ade, J., . . . Mohr, M. (2013). Match performance and physical capacity of players in the top three competitive standards of English professional soccer. Human movement science, 32(4), 808-821.
- Chmura, P., Konefał, M., Chmura, J., Kowalczuk, E., Zając, T., Rokita, A., & Andrzejewski, M. (2018). Match outcome and running performance in different intensity ranges among elite soccer players. Biology of sport, 35(2), 197.
- García-Aliaga, A., Marquina, M., Cordón-Carmona, A., Sillero-Quintana, M., de la Rubia, A., & Refoyo Román, I. (2021). Comparative Analysis of Soccer Performance Intensity of the Pre–Post-Lockdown COVID-19 in LaLiga™. International journal of environmental research and public health, 18(7), 3685.
- Modric, T., Versic, S., & Sekulic, D. (2020). Position Specific Running Performances in Professional Football (Soccer): Influence of Different Tactical Formations. Sports, 8(12), 161.
- Modric, T., Versic, S., & Sekulic, D. (2021). Does aerobic performance define match running performance among professional soccer players? A position-specific analysis. Research in Sports Medicine, 1-13.
- Radzimiński, Ł., Padrón-Cabo, A., Konefał, M., Chmura, P., Szwarc, A., & Jastrzębski, Z. (2021). The Influence of COVID-19 Pandemic Lockdown on the Physical Performance of Professional Soccer Players: An Example of German and Polish Leagues. International journal of environmental research and public health, 18(16), 8796.
- Sæterbakken, A., Haug, V., Fransson, D., Grendstad, H. N., Gundersen, H. S., Moe, V. F., . . . Andersen, V. (2019). Match running performance on three different competitive standards in Norwegian soccer. Sports medicine international open, 3(3), E82.
Staying at your disposal.
Authors
Reviewer 2 Report
The article focuses on changes in match running performance before and after the lockdown. The result looks clear and would be helpful for practitioners. However, some issues should be clarified to verify the changes.
Major Comments
- This is not a randomized controlled study, and impossible to set the control group. Instead, the authors should discuss and speculate about other possible effects on the change in match performance, such as the season/temperature and win/loss. Would you please describe and discuss any potential factors?
- Position-specific performance change is one of the novelties of the study. However, the characteristics of the match performance in each position were not discussed. For example, why match performances are different among positions and why these characteristics changed differently in each position. Would you please add this information in the introduction, hypothesis, and discussion?
Minor Comments
- To make it easier for the reviewer to see what we are pointing out, please number the lines on the right side.
- Make square 2 a superscript font.
- "t-test" is the name of the statistical analysis. What you are treating as a "t-test" is a "t-value" or just "t". Write "t = 1.0" or "t-value = 1.0".
- Please delete ", 2021"
- Please add sample sizes in each position. (for example, Central Defencers Before (n=**) / After (n = **))
- Please be consistent with the decimal points (0.80, not 0.8).
- Discussion what are "CMs"? MFs?
Author Response
The article focuses on changes in match running performance before and after the lockdown. The result looks clear and would be helpful for practitioners. However, some issues should be clarified to verify the changes.
RESPONSE: Thank you very much for recognizing potential of our manuscript.
- This is not a randomized controlled study, and impossible to set the control group. Instead, the authors should discuss and speculate about other possible effects on the change in match performance, such as the season/temperature and win/loss. Would you please describe and discuss any potential factors?
RESPONSE: Thank you very much for your comment. We were fully aware about other possible factors that can influence match performance, such as the season/temperature and win/loss (i.e., contextual factors). As you can see, we already emphasized it in section “4.1. Limitations and strengths” of first manuscript’s version.
However, according to your comment (and according of Reviewer 1 who also pointed this issue) we amended our manuscript and briefly discussed this issue. We are of opinion that in-depth discussion may not be suitable since this was not topic of our study. Please find yellow highlighted text at the end of the Discussion which now reads: “Although this study emphasized possible consequences derived from COVID-19 confinement and its implications for professional football players’ performance, some limitations should be considered. The main limitation of this study was the fact that we did not observe contextual factors. Specifically, factors such as match location, match outcome, strength of opponent or team formation have all been demonstrated as important determinants of physical performance in football [8,31,37,38]. Hence, we are fully aware that such factors could have influenced the results of the presented analysis and that consequently conclusions drawn in this study should be cautiously interpreted. Also, it must be emphasized that only one team was observed, and there-fore findings from this study cannot be generalized. However, this is a very common obstacle in studies involving professional and elite players [39,40]. Future studies should address these limitations by including other teams and contextual factors to provide more comprehensive understanding effect of the COVID-19 lockdown on MRP.”
Note: If you are interested for details regarding contextual factors issue in this study, please see our response to Reviewer 1.
- Position-specific performance change is one of the novelties of the study. However, the characteristics of the match performance in each position were not discussed. For example, why match performances are different among positions and why these characteristics changed differently in each position. Would you please add this information in the introduction, hypothesis, and discussion?
RESPONSE: Thank you very much for this suggestion. We agree with you that characteristics of the match performance in each position were not been discussed appropriately. Therefore, we amended our manuscript by adding this issue in the introduction, hypothesis, and discussion.
Introduction part reads: “Previous studies demonstrated that MRP vary according to the different playing positions of the players due to the different tactical roles in the matches [16,17]. Specifically, central midfielders cover the highest overall distance during the matches, while wingers and fullbacks cover the greatest distance in terms of high-intensity [9,18].”
Hypothesis part reads: “Initially, we hypothesized that MRP would be lower after the COVID-19 lockdown, with certain position-specific differences.” (Please see end of Introduction paragraph – highlighted text).
Discussion part reads: “Although these findings clearly indicate reduced match intensity for all playing positions after COVID-19 lockdown, some specificities in changes of MRP should be noted. Most importantly, it seems that MRP changed differently due to the different game duties of players on different playing positions. For example, for CDs the largest differences between MRP before and after COVID-19 lockdown matches were found for distance covered in running zone (i.e., large effect size). Since most of CD’s efforts in the matches are performed in the zone of running (14.4–19.7 km/h) [9], such findings are actually expected. On the other hand, the FBs’ main technical requirements are the number of entries to the third part of the pitch (i.e., pressing) and the number of crosses [29,30], which are usually performed at moderate and higher speeds. Not sur-prisingly, for this playing position (FBs) we evidenced largest differences for running and high intensity running (both large effect sizes). Next, the sprint distance covered is an important determinant of FWs’ overall game performance [15]. Interestingly, largest differences between MRP before and after COVID-19 lockdown matches for FWs were evidenced for high intensity running (large effect size) which includes high speed running (19.8-25.1 km/h) and sprinting (>25.2 km/h). The main role of MFs is to organize the offense by proper ball control and passes, rather than by invasion into the opponent’s area [29]. Therefore, MFs’ game duties are most likely related to the accelerations and decelerations. Similar like in other playing positions, MFs’ MRP that are related to their main game duties (e.g., total accelerations/decelerations and high intensity decelerations) were decreased in matches after COVID-19 lockdown.” (Please see text highlighted in yellow; Discussion section - 3rd paragraph)
Minor Comments
- To make it easier for the reviewer to see what we are pointing out, please number the lines on the right side.
RESPONSE: We are sorry for this. Now we see that in submission system .pdf and .docx files do not include line numbering. Since we used official IJERPH template, we are not sure how/why this happened. Anyway, line numbers are now added in latest version of manuscript.
- Make square 2 a superscript font.
RESPONSE: Amended accordingly.
- "t-test" is the name of the statistical analysis. What you are treating as a "t-test" is a "t-value" or just "t". Write "t = 1.0" or "t-value = 1.0".
RESPONSE: Thank you for this suggestion. We changed “t-test” to the “t-value”.
- Please delete ", 2021"
RESPONSE: Amended accordingly.
- Please add sample sizes in each position. (for example, Central Defencers Before (n=**) / After (n = **))
RESPONSE: Thank you for this suggestion. We added sample size for each position. Text now reads: “A total of 121 MRPs were obtained, and divided according to player position into the four groups: CD = 38, FB = 20, MF = 46 and FW = 16. Position specific MRP were later divided into the two groups, before and after COVID-19 lockdown: CD before = 19 / CD after = 19; FB before = 8 / FB after = 12; MF before = 16 / MF after = 30; FW before = 5 / FW after = 11” (Please see highlighted text, 2nd paragraph of the subsection Participants and Design).
- Please be consistent with the decimal points (0.80, not 0.8).
RESPONSE: Thank you for this suggestion. We checked all decimal points and applied consistent formatting.
- Discussion what are "CMs"? MFs?
RESPONSE: This was typing error. We changed CMs into MFs.
Staying at your disposal.
Authors
Round 2
Reviewer 1 Report
Thank you very much for your good explanation.
I don´t agree with your arguments but you explain very well your ideas and you defend your paper with a right information.
The role of the winger, attacking midfielder and holding midfielder are too different. You must be consider like a different position in the future research.
Good job.